# Green Synthesis of Silver Nanoparticles Using Pecan Nut (*Carya illinoinensis*) Shell Extracts and Evaluation of Their Antimicrobial Activity

**DOI:** 10.3390/antibiotics11091150

**Published:** 2022-08-25

**Authors:** Alberto Antonio Neira-Vielma, Héctor Iván Meléndez-Ortiz, Josué Israel García-López, Saúl Sanchez-Valdes, Mario Alberto Cruz-Hernández, Josefina Guadalupe Rodríguez-González, Sonia Noemí Ramírez-Barrón

**Affiliations:** 1Centro de Estudios e Investigaciones Interdisciplinarios, Universidad Autónoma de Coahuila, Carretera México km 13, Arteaga 25350, México; 2CONACyT-Centro de Investigación en Química Aplicada, Blvd. Enrique Reyna Hermosillo 140, Saltillo 25294, México; 3Centro de Capacitación y Desarrollo en Tecnología de Semillas, Departamento de Fitomejoramiento, Universidad Autónoma Agraria Antonio Narro, Saltillo 25315, México; 4Centro de Investigación en Química Aplicada, Departamento de Procesos de Transformación de Plásticos, Blvd. Enrique Reyna Hermosillo 140, Saltillo 25294, México; 5Departamento de Ciencia y Tecnología de Alimentos, Universidad Autónoma Agraria Antonio Narro, Calzada Antonio Narro 1923, Saltillo 25315, México; 6Departamento de Ciencias Básicas, Universidad Autónoma Agraria Antonio Narro, Calzada Antonio Narro 1923, Saltillo 25315, México

**Keywords:** green synthesis, silver nanoparticles, pecan nut shell, antimicrobial activity, antioxidant capacity, *Carya illinoinensis*

## Abstract

Nowadays, the increase in bacteria resistant to multiple antibiotics has become a real threat to the human health, forcing researchers to develop new strategies. Silver nanoparticles (AgNPs) may be a viable solution to this problem. The green synthesis of AgNPs is considered a green, ecological and low-priced process that provides small and biocompatible nanostructures with antimicrobial activity with a potential application in medicine. In this work, pecan nut shell extracts were analyzed in order to determine their viability for the production of AgNPs. These NPs were synthesized using an extract rich in bioactive molecules, varying the reaction time and silver nitrate (AgNO_3_) concentration. AgNPs production was confirmed by FT-IR, UV-Vis and EDX spectroscopy, while their morphology and size were determined by transmission electron microscopy (TEM) and dynamic light scattering (DLS). The antibacterial activity of AgNPs was evaluated by the agar diffusion method against *Salmonella typhi*, *Staphylococcus aureus* and *Proteus mirabilis*. The results showed that it is possible to obtain nanoparticles from an extract rich in antioxidant molecules with a size between 39.9 and 98.3 nm with a semi-spherical morphology. In addition, it was shown that the reaction time and the concentration of the precursor influence the final nanoparticles size. Antimicrobial tests showed that there is greater antimicrobial inhibition against Gram-negative than Gram-positive microorganisms, obtaining inhibition zone from 0.67 to 5.67 mm.

## 1. Introduction

Silver nanoparticles (AgNPs) have been used in a wide range of applications due to their particular chemical, optical, electronic and magnetic characteristics, which differ completely from those of bulk metals [1]. AgNPs are known to have microbicidal effects, and they do not impose any adverse effects on the human body. The antimicrobial activity of AgNPs is mainly related to silver ions, which have a wide antibacterial spectrum and are a good option against multidrug-resistant bacteria, which are a real threat to the human health.

In recent decades, research on nanoparticles and nanocomposites has generated great interest, especially in finding more effective ways for nanoparticle synthesis [2]. Green synthesis processes reduce the generation of harmful byproducts that damage the environment and allow an efficient resource-saving synthesis. For this reason, green synthesis using plants and microorganisms including bacteria, fungi, algae and their parts have attracted strong attention. Nanoparticles derived from biological materials are known as biogenic nanoparticles, and the involved synthesis process is known as the green synthesis of nanoparticles [3]. In particular, plant extracts can minimize the use of hazardous solvents [4] and reduce costs [5], since they contain secondary metabolites that can function as stabilizing and reducing molecules, such as phenolic acid, flavonoids, alkaloids and other compounds. These compounds are the main responsible for the reduction reaction of ionic metallic nanoparticles [6,7]. The content of antioxidant biomolecules in plant extracts plays an important role in the production of silver nanoparticles, while the evaluation of the antioxidant capacity can provide an accurate idea of how these extracts can be useful for obtaining metallic nanoparticles [8]. Different extracts have been used to synthesize silver nanoparticles, among them the pecan nut (*Carya illinoinensis)* leaf stands out to synthesize silver nanoparticles [9], since the extracts contain a high amount of phenolic compounds [10]; however, the pecan nut shell represents an important agro-industrial desire, in that it contains a high amount of phenolic antioxidant compounds and proanthocyanidins, including vaillic, caffeic and gallic acids, and also contains catechin and tannic acid [11]. Additionally, research has been carried out to determine the influence of factors such as pH, reaction temperature, reaction time, and the precursor agent concentration [12]. Reported results suggest that when using glutathione as a reducing agent, the increase in its concentration decreases the size of the nanoparticles. The size, shape, and extent of nanoparticle synthesis using plant-derived biomaterials are also highly influenced by the length of reaction time in which the suspension medium is incubated [4]. On the other hand, the solution pH plays an important role in plant-mediated nanoparticle biosynthesis. Several reports have indicated that the pH influences the size, shape, and production rate of the synthesized nanoparticles. At the same time, the solution pH also influences the activity of the functional groups in the plant extract/biomass and influences the reduction rate of a metal salt [13]. Reaction temperature is another important factor that influences the size, shape, and production rate of nanoparticles. Similar to pH, the nucleation and growth process increases with increasing temperature, which, in turn, increases the rate of biosynthesis [14]. Finally, one of the most important applications of metallic nanoparticles, especially AgNPs, is in the medicine field, in which these nanoparticles have been successfully used as antimicrobial agents. The antimicrobial activity of nanoparticles against a wide spectrum of Gram-positive and Gram-negative bacteria and fungi has been reported [15,16].

Thus, the objective of this work was to obtain an extract rich in reductors biomolecules with high antioxidant capacity from the pecan nut shell to ensure the formation of AgNPs. For this reason, these extracts were evaluated for the amount of protein, carbohydrates and antioxidant capacity by 2,2-Diphenyl-1-picrylhydrazyl (DPPH), 2,2′-azino-bis (3-ethylbenzothiazoline-6-sulfonic acid (ABTS), and Ferric reducing antioxidant power (FRAP). Additionally, the reaction parameters such as time and concentration of the precursor agent (AgNO_3_) were evaluated to determine the influence of these variables on the size of the obtained AgNPs. To test the antimicrobial activity of the obtained nanoparticles, they were evaluated against three different pathogenic bacteria: *Salmonella typhi*, *Staphylococcus aureus* and *Proteus mirabilis*. The obtained results are a good option to synthesize AgNPs with antimicrobial characteristics for medical applications using environmentally friendly green synthesis. Compared with other synthetic methods, the synthesis reported in this is work is green, efficient, rapid and simple. The newly prepared AgNPs may have many potential applications such as in chemical, medical and other biological areas.

## 2. Results and Discussion

### 2.1. Protein Determination in Pecan Shell Extracts

The protein analysis performed to the pecan nut shell extract showed values lower than 1% were obtained: 0.17 for 4 h, 0.15 for 8 h and 0.17 for 12 h of reaction (see Table 1). However, by comparing the means with the Tukey test, it is shown that there are no significant differences (*p* < 0.05) between them. Statistically, the protein content is not influenced by the reaction time when preparing the extracts. Western pecan nut shell proteins are often rich in arginine, lysine and threonine [17,18].

### 2.2. Sugar Determination in Pecan Shell Extracts

The concentration of total sugars was determined for each extract. According to the obtained calibration curve, the results are shown in Table 1. It can be observed that as the reaction time increases, the sugar concentrations in samples decreases. It is possible that the extract exposure for a longer reaction time at 80 °C, results in sugar degradation [19] and present Maillard reactions [20]. With the ANOVA analysis, it was possible to determine that increasing the reaction time in the extracts does not significantly increase their sugar content.

### 2.3. Antioxidant Capacity (ABTS, DPPH and FRAP) of Pecan Shell Extracts

The results shown in Table 1 were expressed in equivalent micromoles of Trolox (TE)/100 g. According to the results obtained for the three methods, it is possible to confirm that the extract obtained in the reaction with the shortest time (4 h) is the one with the highest antioxidant potential. Villarreal et al. (2007) [21] reported that pecan nut shells contain a high antioxidant capacity; likewise, De la Rosa et al. (2011) [22] determined phenolic compounds, flavonoids, and proanthocyanidins in pecan nut shells. Similarly, it was observed that, by increasing the extraction time, the antioxidant activity decreases. It is well-known that phenolic compounds, responsible for the antioxidant capacity, are sensitive to adverse environmental conditions, including: temperature, light, pH, moisture, and oxygen [23]. Therefore, they are prone to degrade during processing and certain reaction conditions.

Through statistical analysis, it is shown that significant differences are found between the ABTS and DPPH tests, and between ABTS and FRAP (*p* < 0.05) during the used reaction times. According to the results obtained, it was determined that the extract obtained after 4 h of reaction contains more molecules with reducing capacity. For this reason, this sample extract was chosen for the AgNP synthesis. Therefore, it can be observed that the examined extracts are sources of specific biomolecules with antioxidant potential. This suggested that all the complex structures of these biomolecules contribute to the process of AgNPs formation [14].

When increasing the reaction time to obtain the aqueous extracts, there were no significant differences in terms of the proteins and sugars content, only in the antioxidant capacity. These results suggest that the optimal condition for an efficient extraction of reducing molecules appears to be at least 4 h of reaction time.

### 2.4. Synthesis of AgNPs Using Pecan Nut Shell Extracts

The reaction of the aqueous extract of the pecan nut shell, with different concentrations of silver nitrate, resulted in brown-yellowish solutions, as can be seen in Figure 1. The formation of the nanoparticles is clearly seen due to the change in the solution color before and after the reaction. This reaction resulted in a slight yellow coloration before the reaction, and a dark brown color after it, as can be seen in Figure 1.

Vijayakumar et al. (2013) [24] reported that the reduction in AgNPs during exposure to aqueous extract resulted in a gradual increase in light brown to yellowish brown color change. This change in color is attributed to the nanoparticles electrons collective excitation, which is known as localized surface plasmon resonance. These electrons’ excitation can absorb certain frequencies of incident light and transmit non-absorbed frequencies associated with a certain color [25].

### 2.5. Characterization of AgNPs

#### 2.5.1. UV-Visible Spectrophotometry

The results shown in Figure 2 confirms the presence of AgNPs at a spectrum between 425 and 490 nm. Figure 2 shows the UV spectra corresponding to the wavelength and absorbance obtained for each sample. A wavelength peak can be seen between 411 and 429 nm, which confirms the AgNPs formation.

It can be observed that all the samples contain AgNPs, since all the samples show a peak within the characteristic wavelength range associated with this type of nanoparticle. Bharti (2020) [26] reported that, in the case of silver nanoparticles, the plasmon peak appears at a wavelength around 400–450 nm, and its exact position depends on the diameter, shape and size distribution of the nanoparticles.

#### 2.5.2. Transmission Electron Microscopy (TEM)

The micrographs obtained from the nanoparticles show that they have a semi-spherical shape morphology (Figure 3). Evidence also confirms that the nanoparticles obtained at a higher concentration of AgNO_3_ presented greater particle aggregation. It is clearly observed that, by increasing the silver nitrate concentration, the particle size decreases, which agrees with the results obtained by dynamic light scattering (DLS).

#### 2.5.3. Energy Dispersive X-ray Analysis (EDX)

EDX analysis reveals the elemental composition of the nanoparticle synthesis, which suggests that silver is the primary constituent of the obtained particles (Figure 4). The absorption peak close to 3 keV, which is due to the surface plasmon resonance [25,27]. The peaks 22 keV and 25 keV correspond to the binding energies of AgL, Ag Ka_1_ and Ag Ka_2_, respectively, while the peaks situated at binding energies of 8.0 and 9.01 keV belong to CuKa and CuKb, respectively. Additionally, a signal around 0.7 keV corresponding of carbon is observed. The copper peaks and carbon correspond to the TEM holding grid [28]. However, C and Si signals may be related to the other compounds in the extract [29]. This indicates that the reduction of silver ions to elemental silver has been achieved, which supports the results obtained by UV-VIS.

#### 2.5.4. Fourier-Transform Infrared Spectroscopy (FT-IR)

Figure 5 shows the signals obtained in the FT-IR spectra of the aqueous extract of the pecan nutshell (a) and the AgNPs synthesized from this extract (b). The broad band of 3249–3268 cm^−1^ belongs to the O-H bond stretch, which is characteristic of the phenols [30] present in the extract and the AgNPs. The 1709 cm^−1^ band, which is found in the AgNPs spectrum, can be attributed to the carbonyl groups, which proved the presence of flavanones or terpenoids that are adsorbed on the surface of metal nanoparticles by interaction through π-electrons in the carbonyl groups in the absence of sufficient concentration of chelating agents [31]. The 1612–1604 cm^−1^ peaks correspond to the N-H and C=O stretches coming from the amines and amides present in the proteins found in the extract [9]. The bands of 1110–1007 cm^−1^ correspond to the phenols and polyphenols present in the *Carya illinoinensis* shell extract [9,32]. These results suggest that nanoparticles have a coating on their surface of biomolecules that stabilize them. The shift to 3412 cm^−1^ is related to the breaking of the hydrogen bond, which plays a role in the depletion of silver ions in silver nanoparticles. The nanoparticles consist of the extract compounds that exist in a shell around the nanoparticles due to the similarity of the spectral pattern for the plant extract and the nanoparticles [9].

#### 2.5.5. DLS

Table 2 shows the average diameters of the nanoparticles obtained under different reaction conditions. The average size of the obtained nanoparticles is between 39.9 and 98.3 nm. It was observed that by increasing the concentration of AgNO_3_ and the reaction time, the size of the nanoparticles decreases, obtaining a size of 39.9 nm to 1 mM/12 h of reaction. The difference in the size of the NPs is due to the molar ratio of the extract and the silver nitrate. In previous studies, it has been found that there is an optimal equimolar concentration of AgNO_3_ and extract; when there is an excess of precursor, the particles tend to be larger, while the closer to the optimal concentration of precursor, the nanoparticles they tend to be smaller and narrower distributions [33].

McGinty (2020) [34] attribute this to a particle growth by secondary nucleation that is characteristic when the reactants are consumed and the supersaturation conditions decrease towards equilibrium, which facilitate the growth of the nucleus size and the small particles grow rapidly via this process. That is, at a lower concentration (1 mM) of AgNO_3_, this reagent is quickly consumed, and the nanoparticles will be larger due to secondary nucleation growth. With the increasing reaction time, the rate of the reaction becomes slow because of the protection of stabilizer biomolecules extract, making the accretion of particles size inconspicuous. After this stage, the AgNPs could dissociate due to heating to form smaller particles stabilized by the amine pendant groups on extracts, which leads to the formation of extract-stabilized stable AgNPs [34,35].

### 2.6. Antimicrobial Activity

The average of the measured inhibition zone was obtained from the three repetitions for each treatment and the obtained results can be seen in Table 3 Through a comparison of averages with the Tukey test, it is shown that the significant differences between experiments are found between 1 mM and 4mM (*p* < 0.05), showing statistically that, by increasing the concentration of precursor, there is greater antimicrobial activity. On the other hand, by comparing the means in the same way with the Tukey test for the variable type of bacteria, significant differences were found between the three types of bacteria, demonstrating that the type of bacteria also influences the antimicrobial activity.

The results show better antimicrobial activity against Gram-negative *S. typhi* and *P*. *mirabilis* bacterium obtaining inhibition halos between 1.67 and 5.67mm (see Figure 6). This is in agreement with the reported results from other authors, who note that there is a higher inhibitory effect of AgNPs and Ag+ ions against the Gram-negative bacteria because of the thicker peptidoglycan layer of Gram-positive bacteria, which can obstruct the action of Ag+ ions [36,37]. Other parameters involved in antimicrobial activity are cell physiology, cell metabolism or the degree of contact [38]. In addition, it can be seen that no inhibition was obtained in the experiments at 1 and 2 mM concentrations against *S. aureus*, which highlights that the AgNPs concentration is not suitable to inhibit the growth of this type of bacterium. The best results for Gram-negative bacteria were obtained with the experiments with 2 mM and 4 mM, that is, at higher AgNO_3_ concentrations. The parameters involved in the antibacterial activity of AgNPs are a combination of physical and chemical characteristics, such as the size, shape and surface-to-volume ratio of the nanoparticles, as well as their synthesis method [39]. Silver nanoparticles can adhere to the cell membrane, altering their permeability and respiratory functions. It is possible that such nanoparticles not only interact with the membrane surface but also penetrate into the interior of the bacterium [40,41], triggering DNA fragmentation [42].

## 3. Materials and Methods

### 3.1. Materials

Pecan nut (*Carya illinoinensis*) shell was collected from Parras de la Fuente, Coahuila at the coordinates 102°11′10″ west longitude and 25°26′27″ north latitude, at a height of 1520 m above sea level. Silver nitrate (AgNO_3_) 99.5% was purchased from Fermont company. The *Salmonella typhi*, *Staphylococcus aureus* and *Proteus mirabilis* bacterial culture were previously isolate from Hospital General of Saltillo, Coahuila.

### 3.2. Pecan Nut Shell Extract Preparation

Pecan nut shells were obtained by manual separation of the seed. The shells were then ground in a ball mill (Fritsch, Pulverisette 6 model, Idar-Oberstein, Germany) to obtain a fine powder. This powder was sieved to 250 µm using Tyler sieves. The different aqueous dispersions were prepared as follows: in a ball flask, 1 g of shell powder was suspended in 200 mL of distilled water (pH 6.5) and heated at 80 °C for 4, 8 and 12 h (E1, E2 and E3) under magnetic stirring. To obtain the aqueous extract of *Carya illinoinensis* pecan nut shell, the dispersions were then filtered using a Whatman No.4 filter, and then the protein content was determined by the Kjeldahl method, and the total sugars by the phenol-acid method [43].

### 3.3. Antioxidant Capacity

The antioxidant capacity for DPPH, ABTS, and FRAP were carried out according to López-Contreras et al. [44]. The results were reported in micro-mols of Trolox (6-hydroxy-2,5,7,8-tetramethylchroman-2-carboxylic acid) equivalents per hundred grams of sample (µmol TE/100 g), using as reference the calibration curve of Trolox (0 to 500 µmol/L). All the regents were acquired in Sigma Aldrich ^®^ (St. Louis, MO, USA).

### 3.4. Green Synthesis of AgNPs

The extract with the highest amount of reducing compounds was selected and used for the green synthesis of AgNPs. To determine the influence of stirring time and AgNO_3_ concentration, experiments were performed using the previously obtained pecan nut shell extract at a AgNO_3_ concentration of 1mM, 2mM and 4mM. The AgNPs were prepared using 100 mL of AgNO_3_ solution, which was placed in a ball flask equipped with a condenser and heated to 80 °C with constant stirring for 4, 8, and 12 h. 

### 3.5. Nanoparticles Characterization

#### 3.5.1. UV-Visible Spectroscopic

UV-visible spectroscopy analysis of AgNPs was performed using UV-visible instrument (GBC Scientific, Model Cintra 2020, Hampshire, IL, USA, absorbance mode) in 200–600 nm wavelength.

#### 3.5.2. FT-IR Spectroscopy

Fourier transform infrared (FTIR) spectral measurements were carried out to identify the potential biomolecules in pecan nutshell extract which is responsible for reducing and capping the bioreduced silver nanoparticles. Fourier transform infrared (FT-IR) spectra for *Carya illinoinensis* shell extract powder and silver nanoparticles were recorded within the range of 4000–500 cm^−1^ by using a Nicolet iS50 FTIR spectrometer, Waltham, MA, USA, by KBr pellet.

#### 3.5.3. Microscopy TEM 

AgNPs morphology and size examination were performed using TEM. Briefly, in TEM, a drop of particles suspension was placed on copper film grid and photographed after drying at room temperature using a transmission electron microscopy (FEI-Titan 80–300 kV (Cs = 1.25 mm)), integrated with Energy Dispersive X-ray (EDX) analyzer.

#### 3.5.4. DLS

Dynamic Light Scattering particle size analysis: Hydrodynamic radius sample analysis was recorded at least three times each by the DLS (dynamic light scattering) and electrophoretic light, respectively, methods using a Zetasizer Nano Malvern S90 WR, Malvern, UK. Distillated water (pH 6.5) at a 25 °C was used as the dispersant.

### 3.6. Antimicrobial Activity

The AgNPs antimicrobial effect was determined using filter paper discs impregnated with 100 µL of the AgNPs solution and dried at room temperature. The inoculum of each bacterium was adjusted to 0.5 McFarland, which corresponds to 1.5 × 10^8^ CFU/mL by spectrophotometry. The bacteria evaluated in this determination were *S. aureus*, *S. typhi* and *P. mirabilis*. To determine the bacterial sensitivity, the bacterial suspension previously adjusted to 0.5 McFarland was spread on the surface of the blood agar plate for *S. aureus* and Mueller Hinton (MH) agar for the remaining strains using a cotton swab. Negative controls (C−) were also placed, which correspond to filter paper discs without nanoparticles. The discs impregnated with AgNPs were deposited on the inoculated agar, with at least 3 cm of separation from each other and no more than 5 discs per agar plate, as marked by the CLSI (Clinical and Laboratory Standards Institute). This test was performed in triplicate in a sterile environment. The agar plates were then incubated in an aerobic environment at 37 °C for 18 to 24 h. After 24 h, the inhibition halos were recorded and measured for each system.

### 3.7. Statistical Analysis

All experiments were carried out in triplicate and the degree of significance was analyzed by variance analysis (ANOVA) at a level of *p* < 0.05.

## 4. Conclusions

It is possible to obtain semi-spherical AgNPs using an “ecological” green technique from aqueous extracts of pecan shells with an average diameter between 40 and 98.3 nm, which depended on the initial precursor concentration and the reaction time. Additionally, it was found that these shells contain polyphenols, proteins, and carbohydrates, which are biomolecules responsible for the reduction of silver. Likewise, the obtained AgNPs demonstrated an excellent behavior for inhibiting the growth of two of the three bacteria tested: *Salmonella typhi* and *Proteus mirabilis*. It was demonstrated that, at a higher precursor concentration, the antimicrobial activity is greater. The obtained results are a good option to synthesize AgNPs with antimicrobial characteristics for medical applications using environmentally friendly green synthesis. This also suggest that the AgNPs obtained in this work turn out to be an efficient alternative to design antimicrobial materials.

## Figures and Tables

**Figure 1 antibiotics-11-01150-f001:**
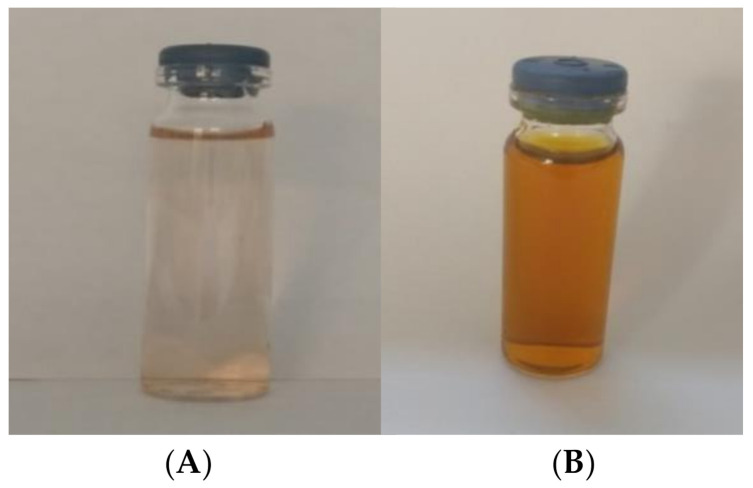
Color of the solution pecan nut shell extract before (**A**) and after (**B**) of reaction with AgNO_3_.

**Figure 2 antibiotics-11-01150-f002:**
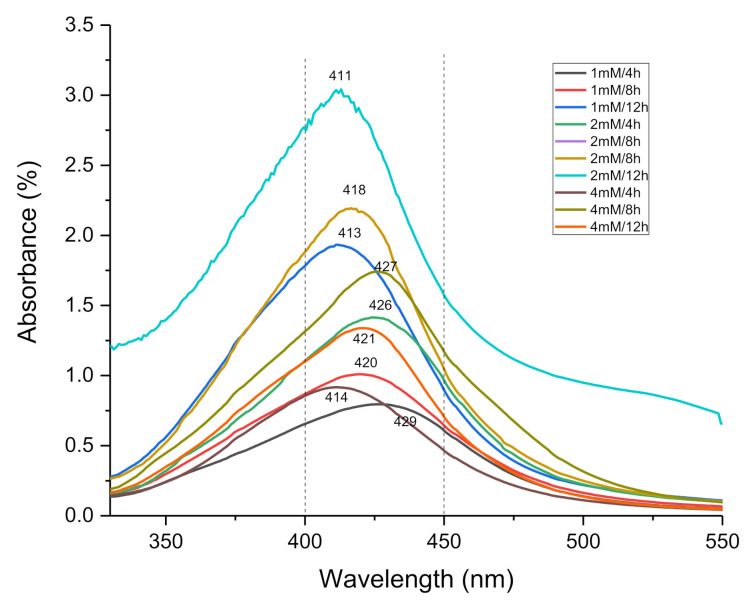
UV-VIS absorbance spectrum of the AgNPs synthetized by pecan nut shell extract at different conditions.

**Figure 3 antibiotics-11-01150-f003:**
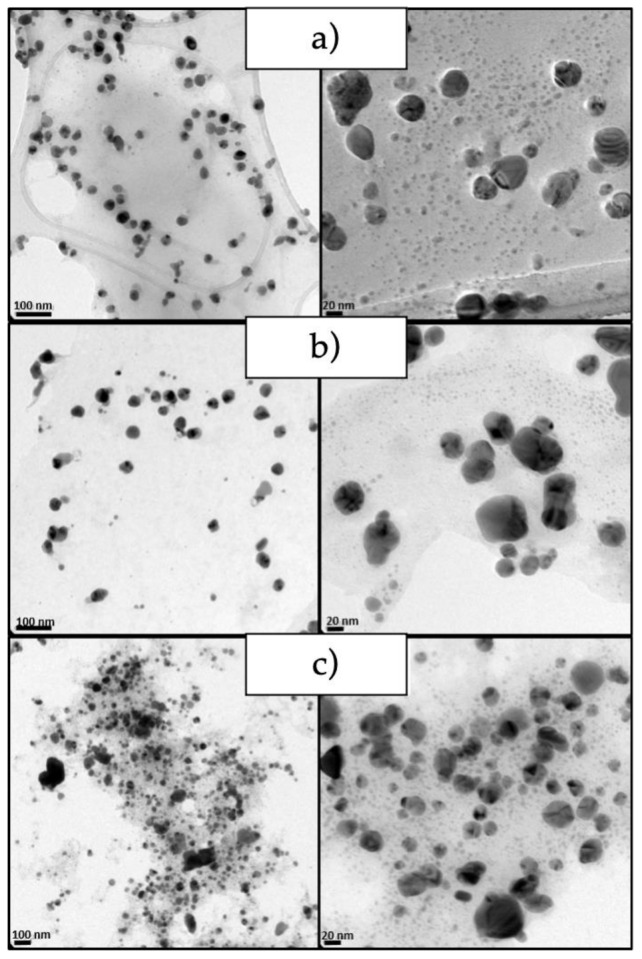
TEM micrograph of the synthesized AgNPs at different reaction conditions (**a**) AgNO_3_ 1 mM/8 h (**b**) AgNO_3_ 2 mM/8 h (**c**) AgNO_3_ 4 mM/8 h. Scale of 100 nm (**left**) and 20 nm (**right**).

**Figure 4 antibiotics-11-01150-f004:**
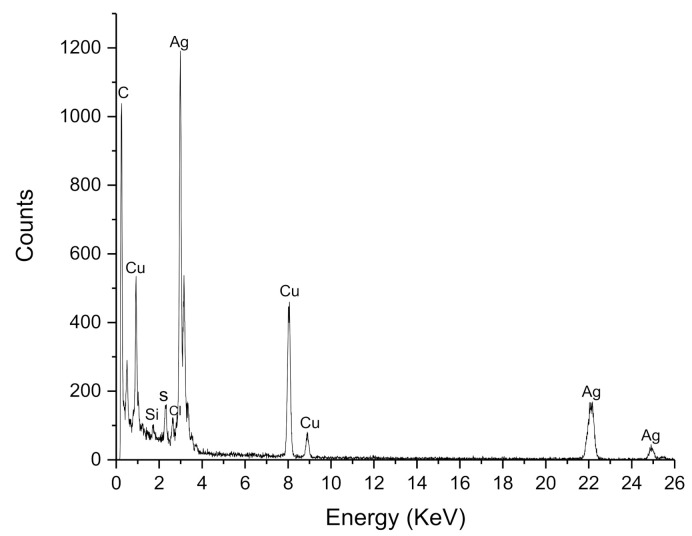
EDX spectrum of AgNPs synthetized by pecan nut shell extract.

**Figure 5 antibiotics-11-01150-f005:**
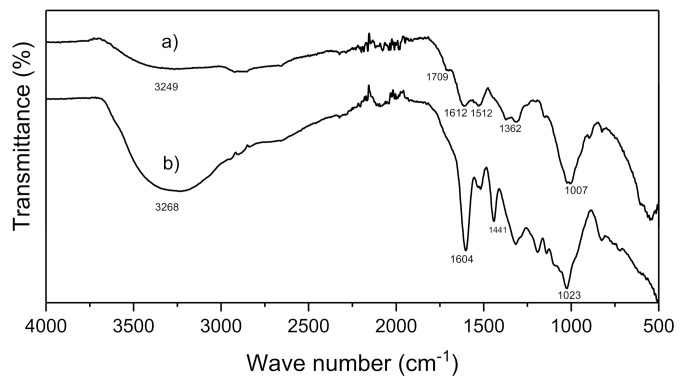
FT-IR of (**a**) silver nanoparticles and (**b**) pecan nut sell extract.

**Figure 6 antibiotics-11-01150-f006:**
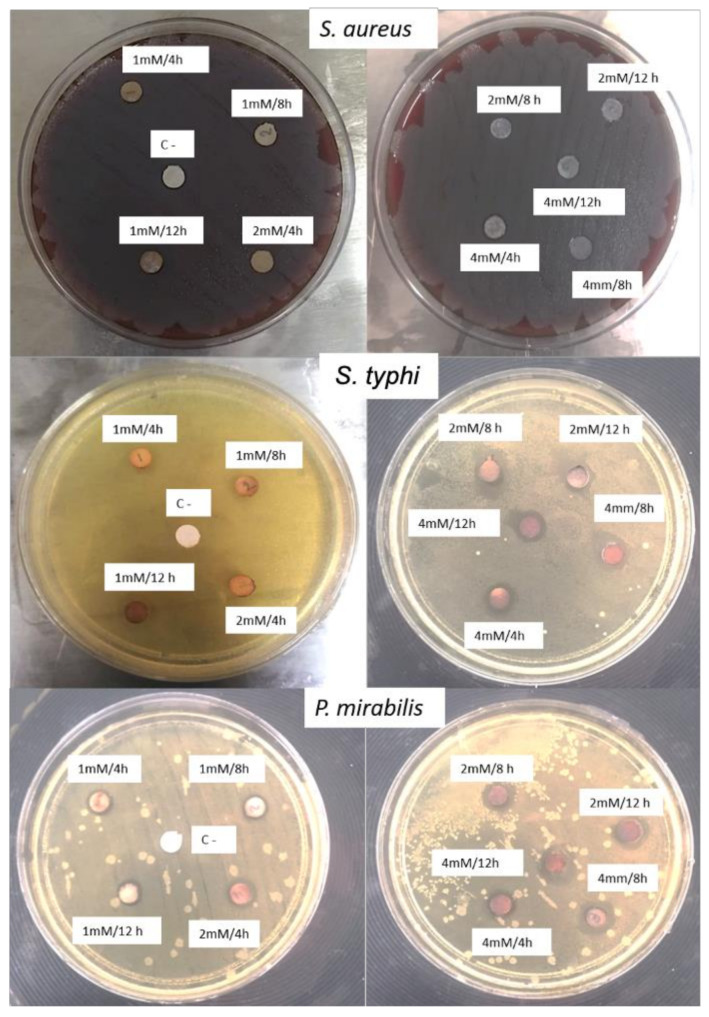
Inhibition zone of *S. typhi* bacteria in Mueller Hinton Agar with AgNPs obtained at different reaction conditions.

**Table 1 antibiotics-11-01150-t001:** Protein, carbohydrates content and antioxidant capacity of pecan nut shell extracts.

Sample	Proteins (%)	Carbohydrates (mg/g of Shell)	ABTS	DPPH	FRAP
			Antioxidant Capacity (µmol TE/100 g)		
E1 (4 h)	0.177 a	352.2 a	1262.6 b	154.9 b	322.4 b
E2 (8 h)	0.159 a	337.6 a	1095.6 a	99.7 a	107.4 a
E3 (12 h)	0.177 a	334.2 a	986.6 a	116.5 a	158.1 a

Means with a common letter are not significantly different between treatments (*p* < 0.5).

**Table 2 antibiotics-11-01150-t002:** Average diameter of the obtained nanoparticles using DLS.

Reaction Time	AgNO_3_ Concentration		
	1 mM	2 mM	4 mM
	Size (nm)		
4	98.3 ± 5.3	78.6 ± 3.7	50.9 ± 3.8
8	93 ± 3.5	44.2 ± 1.7	50 ± 2.6
12	78 ± 1.7	41.6 ± 6.2	39.9 ± 3.1

**Table 3 antibiotics-11-01150-t003:** Inhibition zone of AgNPs synthetized by pecan nutshell extract to different reaction conditions against *S. aureus*, *S. typhi* and *P. mirabilis*.

Reaction Conditions	Inhibition Zone (mm)		
	Microorganism		
	*S. aureus*	*S. typhi*	*P. mirabilis*
1 mM/4 h	WI *	3.17 ± 0.76 b	1.67 ± 0.57 a
1 mM/8 h	WI *	2.83 ± 0.28 a	1.67 ± 0.57 a
1 mM/12 h	WI *	2.83 ± 0.28 a	1.33 ± 0.20 a
2 mM/4 h	WI *	3.67 ± 0.57 b	3.00 ± 0.20 b
2 mM/8 h	WI *	3.67 ± 0.57 b	3.00 ± 0 b
2 mM/12 h	WI *	5.67 ± 1.52 c	3.33 ± 0.57 b
4 mM/4 h	0.67 ± 0.57 a	5.33 ± 1.52 c	3.33 ± 0.57 b
4 mM/8 h	1.00 ± 0 a	4.67 ± 0.57 c	4.00 ± 0 b
4 mM/12 h	1.00 ± 0 a	4.67 ± 0.57 c	3.67 ± 0.57 b

* Without inhibition. Means with a common letter are not significantly different between treatments (*p* < 0.5).

## Data Availability

All the data are incorporated in the manuscript.

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
