# Peer review of "Green Synthesis of Silver Nanoparticles Using Pecan Nut (Carya illinoinensis) Shell Extracts and Evaluation of Their Antimicrobial Activity"

_antibiotics, 2022, doi:10.3390/antibiotics11091150_

Round 1
Reviewer 1 Report
A.A. Neira-Vielma et al. Reported a novel green synthesis method to produce AgNPs from pecan nutshell extracts (Carya illinoinensis). Characterized the nanoparticles and reported their antimicrobial activity against Gram-positive bacteria (S. aureus) and Gram-negative bacteria (S. typhi and P. mirabilis).
Overall, the work is novel, and the manuscript is professionally written and describes methods well. However, the authors need to improve the results and discussion section for a better appeal to the readers of Antibiotics. The manuscript is suitable for publication after addressing the following concerns.
This work is interesting and adds to the existing knowledge of the AgNP synthesis method of natural origin.
Comments:
Major:
Did you perform the statistics on the zone of inhibitions? For example, the 4Mm_ Comparisons between different bacterium types would be interesting and important. It will improve the manuscript’s discussion. Table 4 212-224
Minor:
Results and discussion:
2.2 104 and 105 lines are there any other reasons for sugar value reduction?
2.5.2 173-174 by increasing the silver nitrate concentration, the particle size decreases,
why is that?
182-184 Report the relative percentages of EDX signals for all detected ions. Cite figure 4 in the ext.
2.5.3 DLS
196 Table 3 reports the size as mean plus SD number of repeats/measurements (n)
Figure 5 authors did a respectable job in reporting the entire distribution however did you check the number and volume distribution of the same? Report them in the supplementary section.
Report Span values for each either in the main text or supplementary section refer https://www.materials-talks.com/d90-d50-d10-and-span-for-dls/
Figure 4 is not cited anywhere in the text. Cite it where it is relevant.
Materials and methods:
235 238 report country names and state/city names for all machines and materials. 241 248-249 264 268
274 what was the pH of the water and what kind of water it is?
Did you measure zeta potential? Why was it not reported? at least the range?
Introduction:
A few important references are missing, cite them and differentiate your study from those reported already.
https://www.sciencedirect.com/science/article/pii/S2405844020304692
https://www.mdpi.com/journal/ijms/special_issues/materials_medical_applications
Title:
Include the species name in the bracket next to the title, if possible, for better accuracy
Example: pecan nutshell extracts (Carya illinoinensis).
Keywords: add 2 more
Reviewer 2 Report
The authors have reported the synthesis of silver nanoparticles using pecan nut shell extracts. I would like to give my observations and comments to be addressed for publication of this manuscript in this journal.
1. The use of abbreviations is not consistent throughout the manuscript which can be improved.
2. Authors have reported the synthesis of silver nanoparticles at different silver precursor solution concentrations and the time of synthesis reaction. However, the authors have not clearly addressed the effects of these parameters on the size and shape of the synthesized nanoparticles e.g., how the concentration and time of reaction affect the shape and size should be elaborated with scientific reasoning to describe optimum conditions for the synthesis. As shown in the UV Visible spectra, the correlation between concentration, time, and absorption maxima should be elaborated with reasoning.
3. The authors have evaluated the antibacterial efficiency of the synthesized nanoparticles qualitatively using a zone of inhibition method which does not clearly indicate the antibacterial potential of the synthesized nanoparticles. To address the antibacterial potential of AgNPs, it is recommended to evaluate quantitatively in order to compare the MIC values with the literature.
4. The TEM images show that there is organic content available in the synthesized NPs that might be the biomolecules on the surface of AgNPs. To identify these biomolecules on the surface of AgNPs, FTIR spectroscopic evaluation is necessary.
5. Furthermore, cytotoxicity evaluation of the AgNPs can be included in this manuscript to make it worth publishing in this prestigious journal.
